# Covalent-supramolecular hybrid polymers as muscle-inspired anisotropic actuators

Stacey M. Chin [1], Christopher V. Synatschke [2], Shuangping Liu[3], Rikkert J. Nap[4,5], Nicholas A. Sather[3], Qifeng Wang[3], Zaida Álvarez[2], Alexandra N. Edelbrock [4], Timmy Fyrner [2], Liam C. Palmer [1,2], Igal Szleifer[1,4,5], Monica Olvera de la Cruz[1,3,6,7] & Samuel I. Stupp [1,2,3,4,8]

Skeletal muscle provides inspiration on how to achieve reversible, macroscopic, anisotropic motion in soft materials. Here we report on the bottom-up design of macroscopic tubes that exhibit anisotropic actuation driven by a thermal stimulus. The tube is built from a hydrogel in which extremely long supramolecular nanofibers are aligned using weak shear forces, followed by radial growth of thermoresponsive polymers from their surfaces. The hierarchically ordered tube exhibits reversible anisotropic actuation with changes in temperature, with much greater contraction perpendicular to the direction of nanofiber alignment. We identify two critical factors for the anisotropic actuation, macroscopic alignment of the supramolecular scaffold and its covalent bonding to polymer chains. Using finite element analysis and molecular calculations, we conclude polymer chain confinement and mechanical reinforcement by rigid supramolecular nanofibers are responsible for the anisotropic actuation. The work reported suggests strategies to create soft active matter with molecularly encoded capacity to perform complex tasks.

[1] Department of Chemistry, Northwestern University, Evanston, IL 60208, USA. [2] Simpson Querrey Institute, Northwestern University, Chicago, IL 60611, USA. [3] Department of Materials Science and Engineering, Northwestern University, Evanston, IL 60208, USA. [4] Department of Biomedical Engineering, Northwestern University, Evanston, IL 60208, USA. [5] Chemistry of Life Processes Institute, Northwestern University, Evanston, IL 60208, USA. [6] Department of Chemical and Biological Engineering, Northwestern University, Evanston, IL 60208, USA. [7] Department of Physics and Astronomy, Northwestern University, Evanston, IL 60208, USA. [8] Department of Medicine, Northwestern University, Chicago, IL 60611, USA. These authors contributed equally: Stacey M. Chin, Christopher V. Synatschke  Correspondence and requests for materials should be addressed to S.I.S. (email: s-stupp@northwestern.edu)

**N**ature creates mechanically useful materials through the bottom-up arrangement of nanoscale building blocks into hierarchically ordered structures that bridge length scales and create macroscopic objects. In a human biceps macroscopic forces are exerted through directional and concerted motion of millions of actuating components[1]. These components are made up of actin and myosin supramolecular polymers linked to a macromolecular scaffold known as titin, which collectively form the sarcomere[2]. The aligned structure of sarcomeres over macroscopic scales provides great inspiration for the development of soft materials that behave as anisotropic actuators. When attempting to replicate the functions of skeletal muscle, polymer gels are extremely promising materials due to their ability to undergo large volumetric changes with a variety of stimuli[3–6]. For example, pH-responsive hydrogels successfully adjusted the focal length of optical lenses with only isotropic swelling behavior[7]. Directional actuation can be achieved in molecularly isotropic materials through the application of complex external forces such as thermal gradients, photo-masking of specific locations, or external fields[8–10]. Alternatively, a directional bias in the material itself can be introduced through the use of processing techniques such as the construction of multilayered polymer sheets with different moduli[11,12], or top-down photopatterning of polymer films[13,14]. In supramolecular materials, it is possible to introduce a directional bias of non-covalent assemblies using external forces[15], but very few examples of using such materials as actuators have been reported. One such rare example is a recent contribution that demonstrated the photo-triggered actuation of hierarchically organized molecular motors in aligned supramolecular gels[16]. Other important examples are liquid crystalline (LC) elastomers, where LC phases can lead to reversible deformation of materials[17,18], and shape memory polymers that can recover previously programmed shapes upon application of thermal stimuli[19,20]. However, these LC and polymeric systems are typically hydrophobic and require high temperature processing to create specific shapes. A different approach to anisotropic actuation has been the use of organic-inorganic composite materials to introduce directionality in the microstructure[21]. In one example, anisotropic actuation was achieved by incorporating titania nanosheets aligned by a strong, external magnetic field within a thermoresponsive hydrogel[22]. In another example, oriented patterns of 3D-printed cellulose fibers and clay particles in a thermoresponsive polymer matrix could form complex shapes through predictable self-folding[23]. These systems, however, still require complex processing steps to attain the desired actuation. It remains of great interest in materials science to design anisotropic actuators based on intrinsically ordered systems containing supramolecular and covalent polymers that align under weak shear forces, which takes inspiration from the hierarchical organization of proteins in skeletal muscle.

Here we report on the bottom-up molecular design of a macroscopic soft actuator that deforms anisotropically in an aqueous environment without requiring high temperature or high stress processing, which are typically needed in such systems[24]. The material has a hybrid composition consisting of two components, a supramolecular polymer that under very weak shearing forces provides the anisotropic framework, and transfers this anisotropy to the thermoresponsive motion of the covalent polymer linked to the supramolecular skeleton. Consequently, the hierarchical structure formed in the shape of a macroscopic tube exhibits anisotropic actuation reversibly driven by thermal changes. To build the anisotropic skeleton we use water soluble peptide amphiphile (PA) molecules that under certain conditions are thermodynamically driven to form effectively infinite nanoscale fibers[25], resulting in a LC solution that is easily shear aligned to encode directional information[15,26]. These PA

molecules are chemically designed to initiate the polymerization of the thermoresponsive polymer using atom-transfer radical polymerization (ATRP). As described below, the required steps for bottom-up assembly of this muscle-inspired hierarchically ordered actuator utilize easily applied shear using simple benchtop procedures.

## Results

**Synthesis and materials characterization.** Figure 1 shows a schematic representation of the fabrication process used to prepare the hybrid actuator material. We synthesized a PA (**PA1**, Fig. 1a, blue color) that contains a bromoisobutyryl moiety coupled to the ε-amino group of a lysine residue in order to initiate the growth of covalent polymer chains by ATRP[27]. Coassembly of **PA1** with **PA2** (filler) results in the formation of supramolecular nanofibers containing 10 mol% initiator functionalized molecules (Fig. 1b). Thermal annealing was used to lengthen the nanofibers[25] and create an aqueous lyotropic liquid crystal at 1 wt% in buffer (Fig. 1c, Supplementary Figure 2). We found that this liquid crystal can be extensively shear aligned in a tubular mold containing a rotating metal rod in the center (Fig. 1d)[26]. By retracting the central rod, the chamber containing PA is exposed to $CaCl_2$ solution and this in turn causes gelation of the liquid crystal into a tube shaped object. Based on earlier work, we hypothesized that the supramolecular fibers would orient circumferentially in the tube walls (Fig. 1e). Lastly, the tubular hydrogels are then transferred to a polymerization bath containing a methanol/water solution. This solution contains diethylene glycol methyl ether methacrylate (DEGMA) and oligo (ethylene glycol) methyl ether methacrylate (OEGMA$_{500}$) monomers (molar ratio 95:5 DEGMA:OEGMA$_{500}$), $N,N'$-methylenebisacrylamide crosslinker, catalyst, and reducing agent in order to carry out ATRP[28,29] and generate covalent chains grafted from the supramolecular scaffold (Fig. 1f). Due to the controlled nature of ATRP, polymer chains can only grow from the initiator sites on **PA1**. Unless otherwise noted, chains had an expected degree of polymerization of 1500, and crosslinker was added at a concentration equivalent to 1 wt% of the monomers, resulting in approximately one crosslink per 75 monomer units. Crosslinking is expected to occur both within polymer chains on the same nanofiber, as well as between nanofibers.

The random copolymer exhibits the expected lower critical solution temperature (LCST) of 35 °C as determined by dynamic light scattering (DLS, Supplementary Figure 3)[30]. We monitored the consumption of monomer using nuclear magnetic resonance (NMR, Supplementary Figure 3). We estimate the molecular weight of tethered chains by gel permeation chromatography of non-tethered chains grown from soluble initiator (ethyl α-bromoisobutyrate), which has been shown to be in good agreement with those grown from surfaces[31]. While the tube dimensions and fibrous morphology (Supplementary Figure 4) remain the same before and after polymerization, distinct differences in the material opacity are observed due to the crosslinking density (photographic insets, Fig. 1e, f). Further discussion of the material opacity can be found in Supplementary Note 5. The hybrid material was stable in a hydrated state at ambient temperature for more than one year. Furthermore, mass calculations show a decrease of the water content from approximately 98% to 94% after polymerization, as well as an increase of overall dry mass, indicating successful polymerization of the covalent polymer. The hybrid tubes can be easily handled and are mechanically much more robust than tubes formed only by the supramolecular scaffold. Microindentation experiments indicate a fourfold increase in radial compressive modulus for the hybrid tubes relative to their supramolecular counterparts (4.2 ±

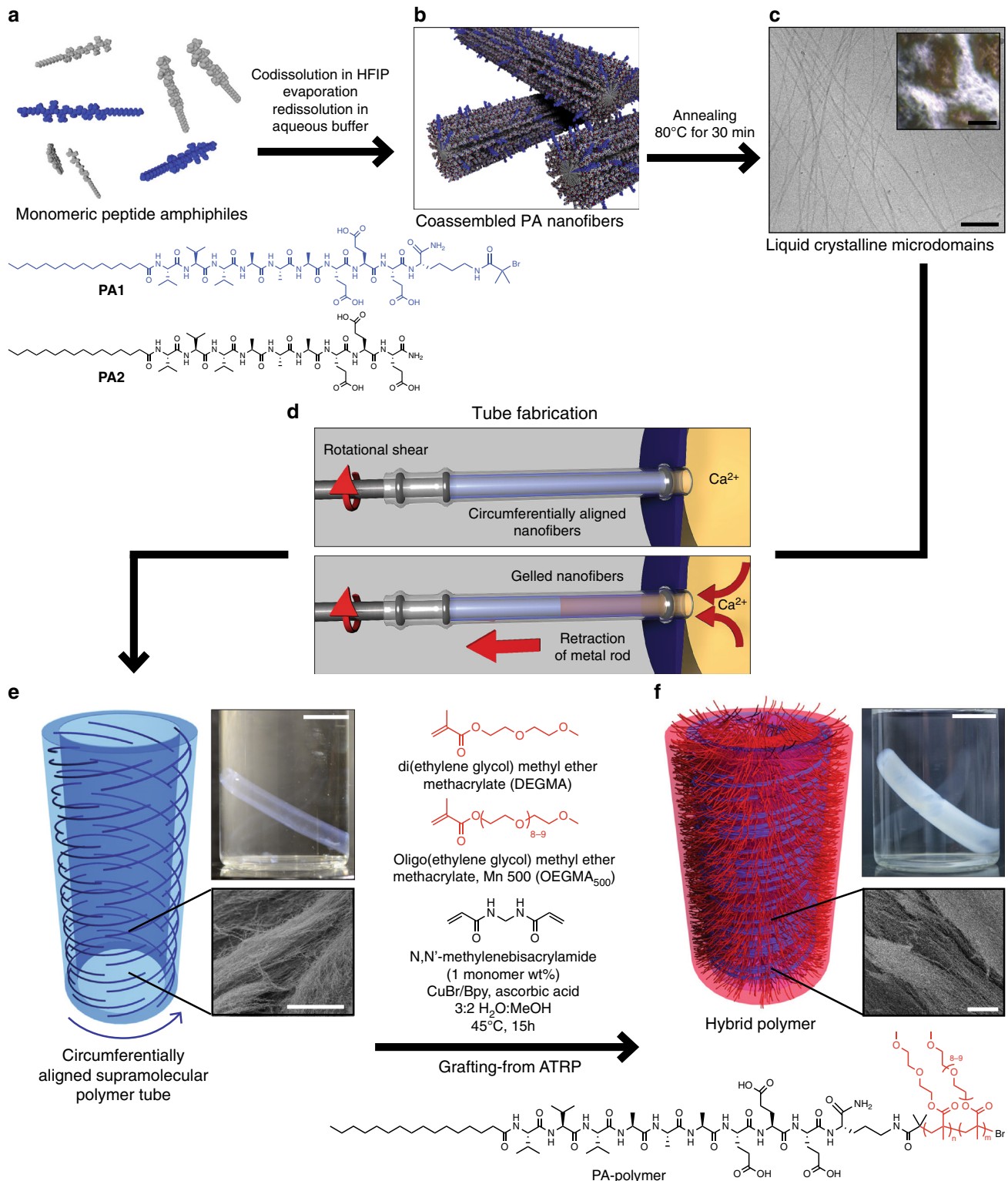

**Fig. 1** Preparation of hybrid actuators. **a** PA1 and **PA2** are codissolved in hexafluoroisopropanol (HFIP) to ensure complete dissolution and molecular mixing. Removal of HFIP and redissolution in aqueous buffer gives **b**, coassembled peptide amphiphile (PA) nanofibers. Subsequent annealing at 80 °C for 30 min gives a LC solution of high aspect-ratio PA nanofibers. **c** Cryogenic transmission electron microscopy shows long nanofibers (scale bar is 200 nm), and polarized optical microscopy (inset, scale bar is 200 μm) shows birefringent domains. **d** To fabricate the PA tubes, rotational shear force is applied to PA solution to circumferentially align the nanofibers within a tubular mold. The central rod is then retracted to allow influx of CaCl₂ solution, gelling the PA as shown in **e**, maintaining nanostructure orientation. Covalent polymer chains are grafted from nanofibers with atom-transfer radical polymerization in a polymerization bath. **f** The resulting hybrid contains covalent chains grafted radially from the nanofiber surface and shows a distinct opacity change from the unpolymerized state (photographic insets, scale bars are 1 cm), while maintaining fibrous morphology (scanning electron microscopy insets, scale bars are 10 μm)

1.5 vs. $16.7 \pm 2.8$ kPa, mean ± s.d., $n = 3–5$). We examined the distribution of covalent polymer throughout the tubular structure using confocal microscopy and fluorescently labeled materials. In both the tube cross-section and the tube wall we observed a homogeneous distribution of fluorophores and a high degree of colocalization (Supplementary Figure 6). This indicates that during polymerization, monomer can freely diffuse within the supramolecular scaffold, allowing polymer chains to be initiated throughout the entire volume of the tube.

**Anisotropic actuation of hybrid materials.** Next, we investigated the actuation behavior of the hybrid tubes by immersing them in water and heating from 25 °C to 70 °C over a period of 30 min. Above the LCST of the copolymer, we observed a significant

contraction of hybrid tubes (Fig. 2a, d). The contraction is reversible and upon cooling, the polymer re-swells to its original dimensions. The hybrids can be actuated for multiple cycles and with similar dimensional changes (Supplementary Figure 8). The actuation of these materials is sufficiently strong to perform work (Supplementary Figure 9), with samples able to lift up to 380 times their dry weight with a work capacity of 0.629 kJ kg$^{-1}$ and volumetric energy density of 5.656 kJ m$^{-3}$.

Hybrid tubes containing circumferentially aligned supramolecular scaffold exhibited anisotropic actuation, contracting more strongly along their length relative to their width (62% versus 79% of original dimensions, see Fig. 2a). This behavior is not observed in tubes containing only covalent polymer or only supramolecular scaffold (Fig. 2b, c). Tubular hydrogels containing

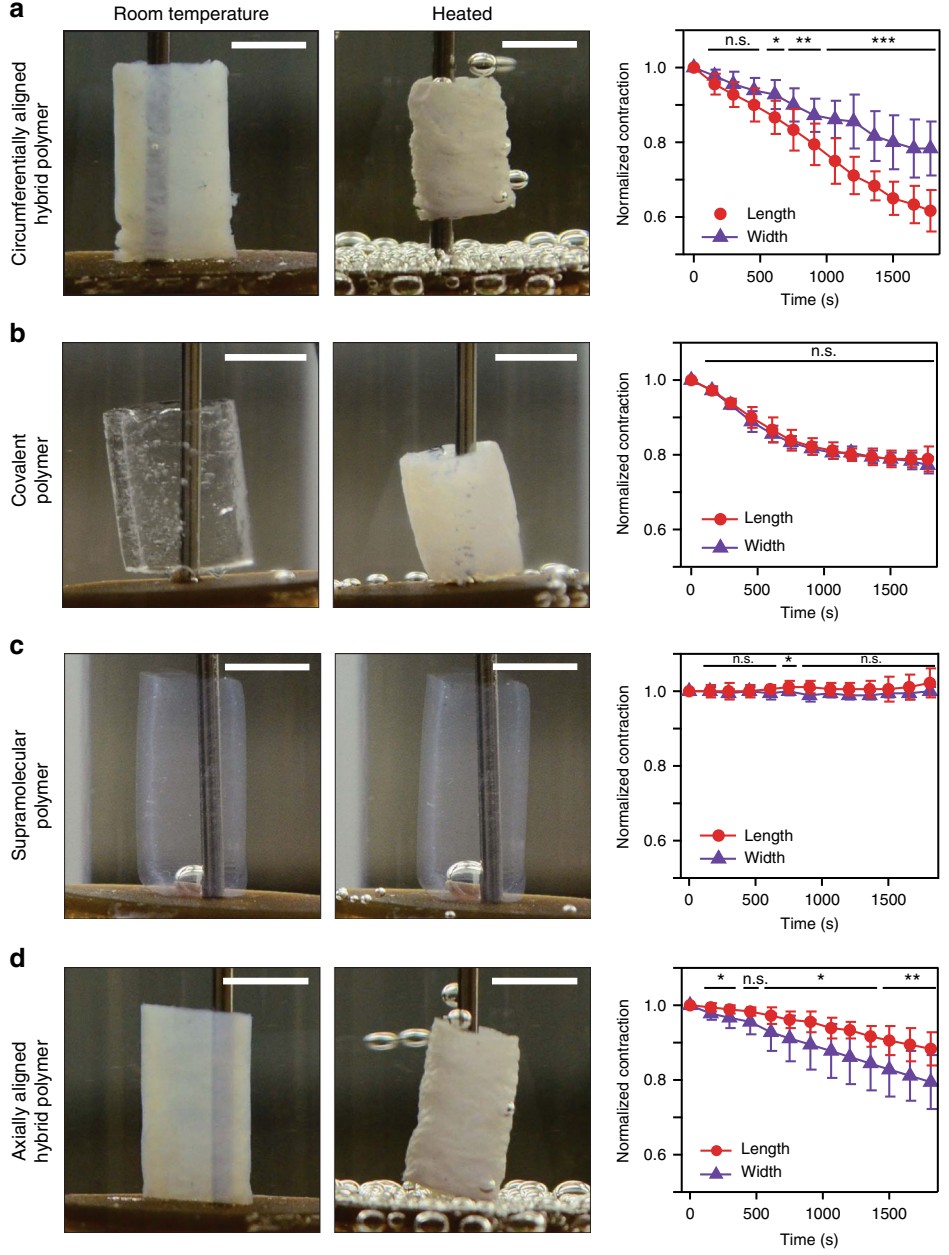

**Fig. 2** Anisotropic actuation in hybrids. **a** Circumferentially aligned hybrid polymer (left) shows anisotropic contraction along the length of the tube upon heating (center), as normalized to original tube dimensions (right) over the course of 1800 s. **b** Covalent polymer tube. **c** Supramolecular polymer tube. **d** Axially aligned hybrid polymer. (scale bars are 3 mm). A representative heating curve is depicted in Supplementary Fig. 7. Statistical analysis was performed using an unpaired two samples Student´s $t$-test; *$p < 0.05$, ** $p < 0.01$, ***$p < 0.001$; (Data are presented as mean ± s.d., $n = 3$)

only covalent polymer contract isotropically to 77% of their original length and width, while supramolecular PA tubes, as expected, have no observable contraction. When we aligned the supramolecular scaffold axially (parallel to the tube long axis) instead of circumferentially (Fig. 2d), the anisotropy reverses to 79% of original width versus 88% of original length. These observations suggest that anisotropic actuation is linked to the alignment of the supramolecular scaffold containing covalently grafted thermoresponsive polymer.

**Alignment of the supramolecular scaffold in hybrid materials.** The data presented in Fig. 2 indicates that anisotropic actuation originates from the combination of covalent polymer and supramolecular phases in the hybrid material. The difference in response between circumferentially and axially aligned polymers

indicates that the direction of alignment in the supramolecular phase determines the nature of the anisotropy. To investigate the importance of this alignment, we analyzed samples with polarized optical microscopy (see Supplementary Figures 10–12). The circumferentially aligned hybrid material shows strong birefringence in both the cross-section and the tube wall, indicating the presence of highly aligned structures (Fig. 3a, d). In contrast, the axially aligned sample only shows strong birefringence in the tube wall but little in the tube cross-section, which is expected for a material with this orientation (Fig. 3b, e). The cross-section and wall of a tube fabricated from covalent polymer hydrogel show negligible birefringence, thus revealing no net orientation (Fig. 3c, f). We also characterized orientation in the tubular samples using small-angle x-ray scattering (SAXS) (see Fig. 3g–i, Supplementary Figure 13). The 2D scattering of aligned samples reveals

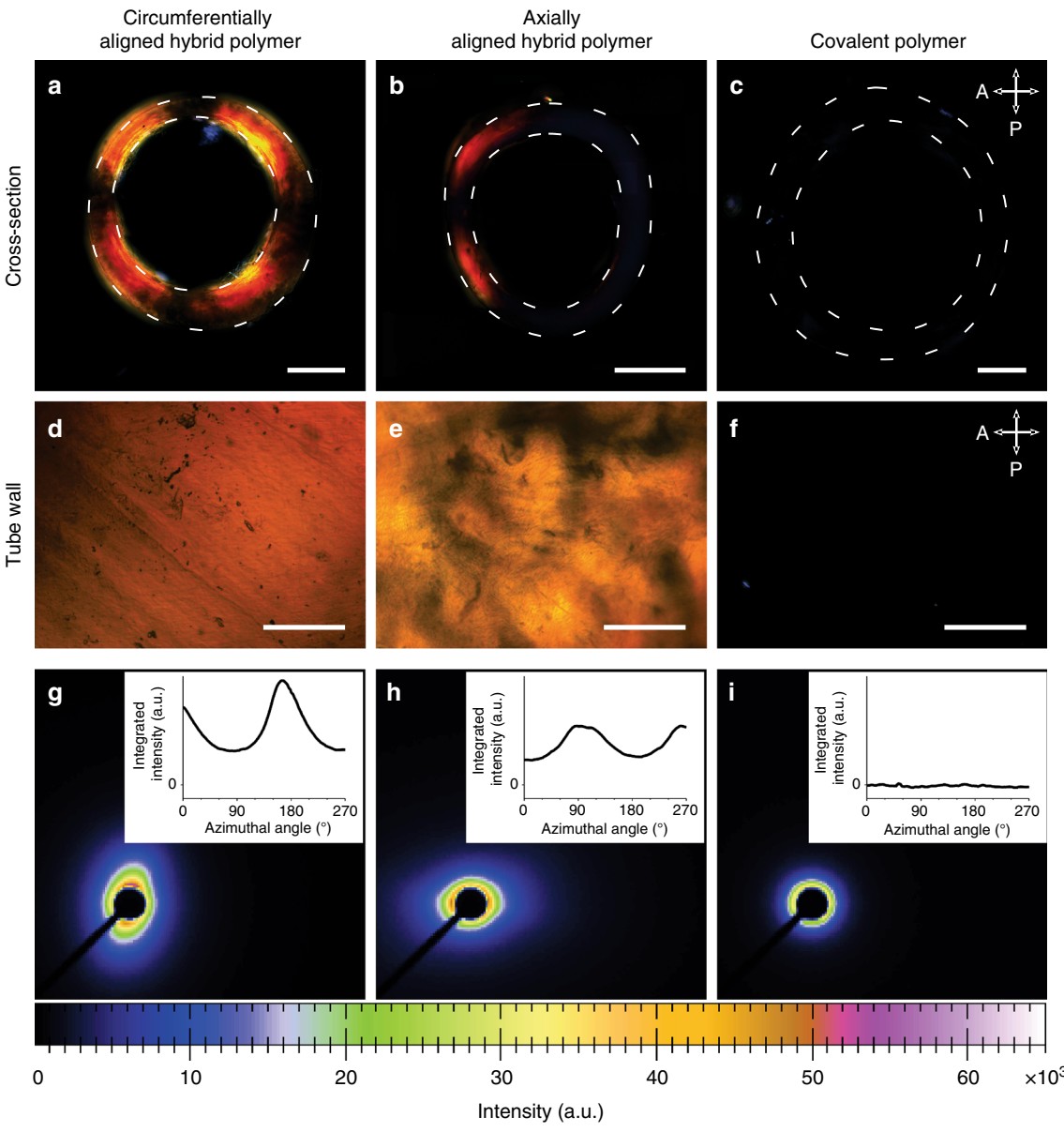

**Fig. 3** PA alignment in hybrid materials. Circumferentially aligned hybrid polymer shows birefringence in cross-sectional slices (**a**) as well as along the tube wall (**d**) under cross-polarized light. Axially aligned hybrid polymer shows birefringence only in the tube wall (**b, e**). Covalent polymer shows negligible birefringence (**c, f**). 2D small-angle x-ray scattering patterns show angle dependent intensity maxima in the aligned hybrids but not in the covalent polymer (**g–i**). Insets show integrated radial intensity versus azimuthal angle. Scale bars in images of cross-sectional slices (**a–c**) are 1 mm and 400 μm in images of the tube walls (**d–f**)

ellipsoidal patterns consistent with long-range order, whereas unoriented ones have no angular dependence of the observed scattering. The plot of integrated radial intensity as a function of azimuthal angle (Fig. 3g, h insets) in the circumferentially and axially aligned hybrid material show maximum intensities around 180° and 90°, respectively, confirming the proposed alignment of supramolecular nanofibers within tubular samples. Finally, the SAXS data reveals that tubes composed of the covalent polymer contain randomly oriented chains (Fig. 3i). It is therefore clear that the strongest contraction of the tube is observed perpendicular to the orientation of supramolecular nanofibers. In the case of circumferentially aligned nanofibers, the tubes shrink preferentially along the length, whereas those with axial alignment reveal preferential shrinkage along the width. We therefore have strong evidence that anisotropic contraction of the tubes is directly linked to the macroscopic orientational order provided by the supramolecular phase.

**Mechanisms of anisotropic actuation**. We considered possible mechanisms that link the observed correlation between anisotropic actuation and orientation of the supramolecular phase. One possibility is that high-persistence length structures mechanically restrict contraction of the covalent polymer. In this case, the oriented, rigid structures would diminish contraction preferentially in the direction of alignment and lead to macroscopic anisotropic actuation (Fig. 4a). Tensile experiments show a difference in the Young's modulus perpendicular and parallel to the nanofiber alignment axis (50.3 ± 39 kPa vs. 263.3 ± 179 kPa, mean ± s.d., $n = 6$–12), indicating the nanofiber orientation does affect the material mechanics (Supplementary Figure 14). We hypothesize that if the anisotropic response were solely due to mechanical restriction from the rigid nanofibers, a composite material–where the polymer is not covalently connected to the

supramolecular nanofibers–should match that of the hybrid materials. We therefore synthesized composite samples containing circumferentially aligned **PA2** nanofibers embedded in a covalent polymer matrix (Fig. 4b), in contrast to the previously described hybrid materials. These composite samples show similar integration of the covalent polymer component throughout the material (Supplementary Figure 6). We found that composite samples do exhibit anisotropic actuation, shrinking to 85% of their original length and 92% of their original width. However, this actuation is of considerably lower magnitude than that of the previously described circumferentially aligned hybrid samples (Fig. 2a). This indicates that while mechanical restriction contributes, it cannot be the sole cause of the observed hybrid anisotropy.

In addition to the preparation of a composite control, we also constructed a computational model using finite element analysis. However, it is challenging to directly model this system due to the vast differences in length scale between supramolecular nanofibers and the macroscopic tube. Therefore, we treat the bulk hybrid tube as a periodic array of rigid, non-deformable PA nanofibers embedded in a soft matrix so that the behavior of the system can be captured using a unit cell with periodic boundary conditions. Specifically, each unit cell contains one PA nanofiber, which is modeled as a cylindrical rod strongly adhered to the surrounding polymer in order to prevent slippage at the PA-polymer interface and therefore constraining the matrix during contraction (Fig. 4c). In this model, the size of the unit cell represents the distance between fibers, and the PA nanofiber is considered an infinitely long object given the periodic boundary conditions. This assumption is a reasonable approximation given the very high aspect ratio and relative stiffness of PA nanofibers (see Supplementary Note 3). Below, we describe this finite element model of a soft matrix containing dispersed, rigid nanofibers that has successfully been used to describe deformed hybrid hydrogels[32,33].

The free energy, $F$, of the covalent polymer in the hybrid structure is described using the Flory–Rehner theory[34], which incorporates the elastic energy of the polymer gel into the free energy of mixing of the solvent and the polymer gel,

$$F = \frac{1}{2}NkT\left[\lambda_x^2 + \lambda_y^2 + \lambda_z^2 - 3 - \ln\left(\lambda_x\lambda_y\lambda_z\right)\right] + kT$$
$$\frac{V_m}{\nu}\left[\left(\frac{1}{\phi} - 1\right)\ln(1 - \phi) + \chi(1 - \phi)\right] \tag{1}$$

where $N$ is the total number of polymer chains (defined as segments between crosslinking junctions), $\lambda_x$, $\lambda_y$, and $\lambda_z$ are the stretching ratios along the principal axes (final length after deformation/initial length), $V_m$ is the volume of the polymer in the absence of solvent, $\nu$ is the volume of each solvent molecule, $\phi$ is the volume fraction of monomers, and $\chi$ is the Flory interaction parameter between the solvent and the polymer. For a polymer gel with LCST behavior, $\chi$ becomes larger when the temperature increases. For simplicity, we assume here that $\chi$ only depends on temperature. In an isotropic gel, the stretching ratios are the same along all principal axes and therefore $\lambda_x = \lambda_y = \lambda_z = \phi^{-1/3}$. In the case of an anisotropic gel, where the gel is constrained uniaxially along the $z$ direction (fiber direction) and free to swell in the $x$ and $y$ directions, $\lambda = \lambda_x = \lambda_y = (\phi\lambda_z)^{-1/2}$. By varying $\chi$, a change in $\phi$ and consequently $\lambda$ is directly obtained from the equilibrium equations, and therefore, the relation between $\chi(T_i)$ and $\chi(T_f)$ and the resulting change in the stretching ratios $\lambda$ can be derived (see Supplementary Note 3),

$$\frac{\chi}{\lambda^2\lambda_z} = \frac{1}{2}\frac{N\nu}{V_m}\left(1 - 2\lambda^2\right) - \lambda^2\lambda_z\ln\left(1 - \frac{1}{\lambda^2\lambda_z}\right) - 1 \tag{2}$$

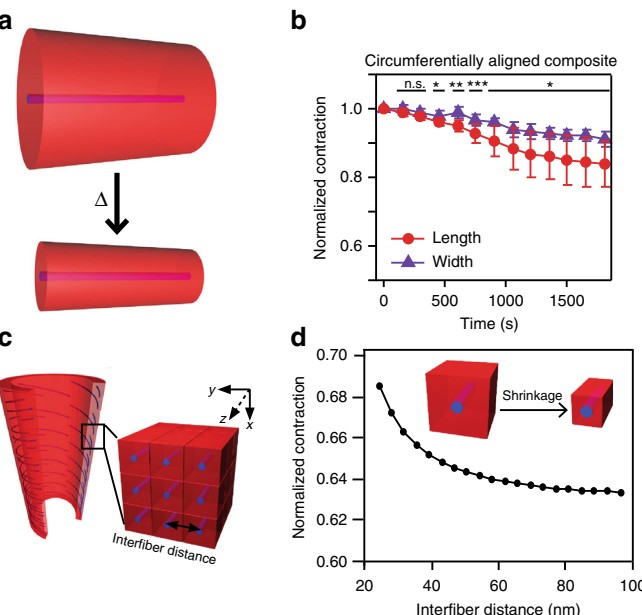

**Fig. 4** Mechanical reinforcement of PA nanofibers. **a** PA nanofibers with high persistence length provide mechanical reinforcement along the PA axis, preventing contraction in the direction of alignment. **b** Experimental results of heating a circumferentially aligned composite material. Statistical analysis was performed using an unpaired two samples Student's t-test; *p < 0.05, **p < 0.01, ***p < 0.001; (Data are presented as mean ± s.d., n = 3). **c** Schematic of finite element analysis model of covalent-noncovalent system. **d** Finite element analysis results of shrinkage due to mechanical reinforcement with varying interfiber distance

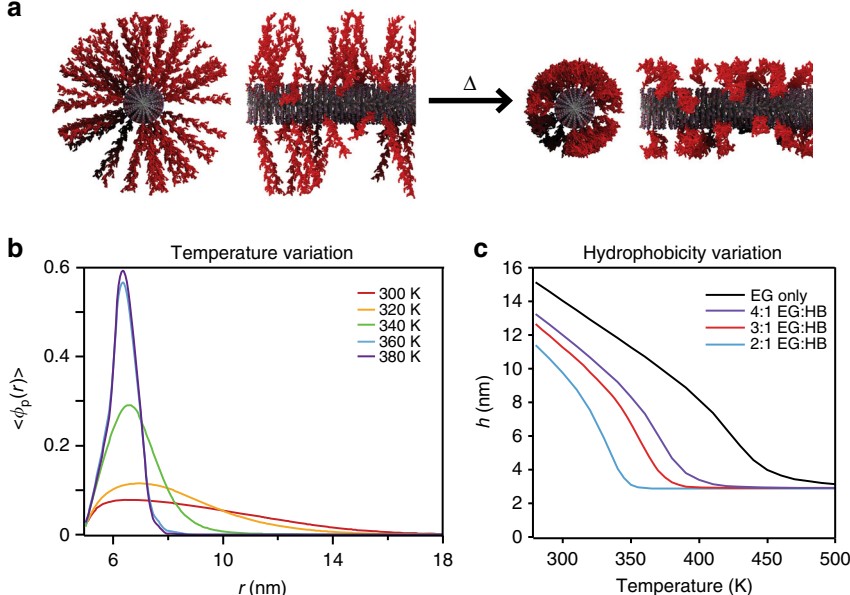

**Fig. 5** Confinement of polymer chains. **a** Schematic representation of confinement effect in grafted polymer chains below (extended) and above (collapsed) the transition temperature leading to pronounced volume changes in the material perpendicular to the supramolecular nanofiber. **b** Polymer volume fraction versus radial distance away from the PA nanofiber. Polymer is a linear alternating copolymer $(EG_2\text{-}a\text{-}HB)_{66}\text{-}EG_2$. **c** Height of end-tethered polymer as function of temperature for different copolymers with increasing amounts of hydrophobic monomers. The polymer is a linear alternating copolymer $(EG_n\text{-}a\text{-}HB_1)_m$. Number of segments is $N_p = 200$, the surface coverage is 0.07 chains $nm^{-2}$, and radius of nanofiber equals 5 nm

Estimates of the interfiber distances range from approximately 30 nm in a two-dimensional crystalline state previously observed[35] to 80 nm in a hydrated gel (see Supplementary Note 3). The finite element analysis indicates that the hybrid material should have significantly larger contraction perpendicular to the fiber axis within the range of interfiber distances modeled from 24 to 96 nm (Fig. 4d). Within this range of interfiber distances, the analysis predicts a contraction ranging from 69 to 63% of its original length. This model likely overestimates the contraction, as it does not reflect all aspects of the hybrid system. In particular, the thermal response of the polymer, as well as the chemical structure and unique molecular architecture of the hybrid is not properly represented in the finite element analysis. To complement the finite element analysis and investigate other mechanisms that would aid in the occurrence of anisotropy, we extended an existing molecular theory that describes the chemistry and molecular architecture of the hybrid more accurately.

It is known that end-grafting of chains into polymer brushes greatly affects the resulting physical properties. At high densities, the grafting of polymer chains to a surface leads to polymer chain confinement resulting in more extended conformations compared to those in solution[36]. We expect that, due to the grafting, the more extended conformations of chains perpendicular to the nanofiber lead to a more pronounced radial collapse of polymer chains above the LCST (Fig. 5a). Since the previously described finite element model does not take this molecular picture into account, we used theory to model a single PA nanofiber with end-grafted polymer chains.

The molecular approach used in this work is based on a theory that has successfully been applied to study structural and thermodynamic properties of a variety of end-tethered polymer systems[37]. The theory explicitly includes conformations of polymer chains, as well as size, shape, and volume of every molecular species in the system. For simplicity, the model represents the thermoresponsive DEGMA/OEGMA$_{500}$ copolymer as a linear poly(ethylene glycol) (PEG)-like copolymer of ethylene

glycol and hydrophobic monomers. This allows us to investigate the effect of end-grafting and molecular chemistry on the structure and thermal response of the system. We assume the polymer chains to be evenly distributed and irreversibly end-tethered to the cylindrical nanofiber surface. Previous work has explored PEG chains end-tethered to planar surfaces[38], as well as the behavior of PEG solutions[39]. It was found that in order to accurately represent the LCST behavior of PEG polymers, hydrogen bonding between water and monomer, as well as between water molecules, must be included. Here, we go well beyond the Flory–Rehner approach employed in the finite element analysis described above. These specific hydrogen bond interactions, in addition to the typical polymer-water and polymer-polymer interactions, are necessary to obtain the correct thermal response. In fact, the hydrogen bonding is explicitly included into the free energy expression of the PA nanofibers (see Supplementary Note 4).

For a given grafting density, the model allowed us to calculate the polymer volume fraction, $\left\langle \phi_p(r) \right\rangle$, at a given distance from the surface, $r$. Upon increasing the temperature above the transition temperature, a rapid collapse of the polymer chains is predicted, accurately replicating the experimentally observed LCST behavior (Fig. 5b). The average height, $h$, of the polymer brush is defined as twice the normalized first moment of the volume fraction, $<r>$, minus the radius of the nanofiber, $R$,

$$h = 2(\langle r \rangle - R)\ \text{with}\ \langle r \rangle = \int_R^\infty r \left\langle \phi_p(r) \right\rangle \mathrm{d}r \Big/ \int_R^\infty \left\langle \phi_p(r) \right\rangle \mathrm{d}r \quad (3)$$

Assuming that every tenth PA molecule grows a polymer chain, we estimate a grafting density of 0.07 chains $nm^{-2}$ on the nanofiber surface[40]. For a copolymer of 200 segments, we observe a drastic change in the average height of the brush. In our model, the height of the brush decreases from approximately 12 to 2.75 nm (23% of original height) when the temperature is increased

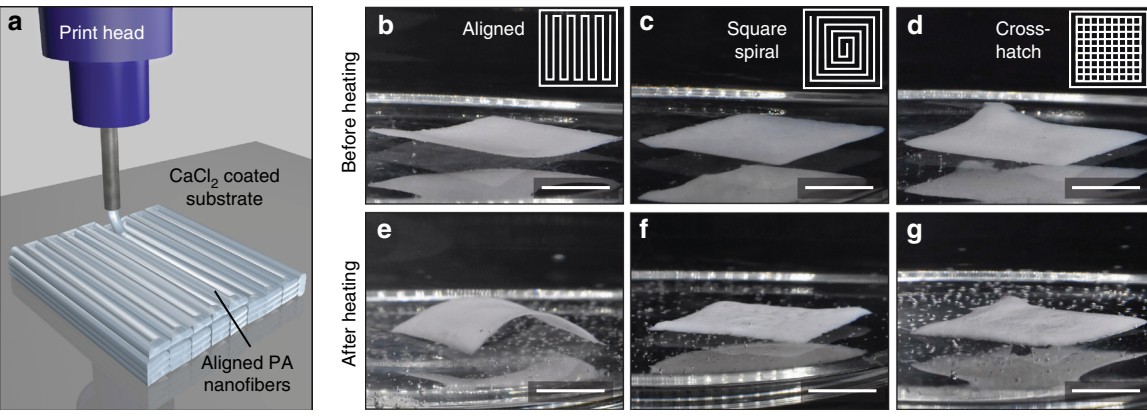

**Fig. 6** 3D printing of hybrid sheets. **a**. Schematic of 3D printing process. Aligned sheets before (**b**) and after (**e**) heating. Square spiral printed sheets before (**c**) and after (**f**) heating. Cross-hatched sheets before (**d**) and after (**g**) heating. Insets show printing patterns of each sheet. (scale bars are 5 mm)

from room temperature to 375 K, and variations in grafting density in the range of that used experimentally result in only minor changes in contraction (Supplementary Figure 16a). This collapse is induced by the relative reduction of polymer-water hydrogen bonds in favor of water-water hydrogen bonds (Supplementary Figure 15). The extent of the collapse predicted by the model is more extreme than what is observed in the experimental system. This is due to the fact that in this calculation the chains are linear and not crosslinked, and are likely more strongly affected by confinement effects. A qualitatively similar thermal response was predicted by the model when we vary grafting density, molecular weight, and chemical composition (Fig. 5c, Supplementary Figure 16). Increasing differences in brush height between the extended and collapsed states were found to occur when grafting densities and molecular weights were increased. We found that the strategy to control the thermal response is to vary the hydrophobic component of the polymer, which can drastically change the transition temperatures (Fig. 5c). This could be and important handle for the design of further actuating materials. In summary, the molecular approach indeed predicts the LCST behavior of the polymer and the connection between anisotropic actuation and the radial collapse of chains in a brush configuration on the nanofiber surface. We conclude that end-grafting of chains on the supramolecular fibers and mechanical reinforcement of the hydrogel by the fibers both contribute to the anisotropic actuation observed in the experimental system.

**3D printed hybrid materials**. Our material can be aligned with weak shear forces, allowing us to use other fabrication techniques such as extrusion printing. 3D printing methods can produce more intricate PA architectures than the previously described tube. We proceeded with the fabrication of sheets from extruded PA filaments as shown in Figure 6a, which are subsequently polymerized in the same manner as the tubes. Because the hybrid sheets contain the same mechanical reinforcement and end-grafting as the tubes, we expect to observe similar anisotropic actuation perpendicular to the direction of supramolecular alignment. Interestingly, sheets that have a net alignment bend in a distinct orientation upon heating (Fig. 6b, e). In contrast, sheets lacking a net alignment, either through the printing of concentric spirals or with a cross-hatch pattern do not show similar bending, but rather non-uniform wrinkling (Fig. 6c, d, f, g). A more detailed analysis of the behavior of these samples can be found in Supplementary Note 6. Our results demonstrate that these aligned hybrid materials are a general platform for programmable

actuation and future work will explore more complex shapes using this 3D printing method.

## Discussion

We have demonstrated the bottom-up molecular design of a soft anisotropic actuator based on conjugation between a supramolecular and a covalent polymer. Directed by the facile alignment and internal stability of mesogenic supramolecular polymers, it was possible to induce anisotropic actuation in an extremely flexible, thermoresponsive covalent polymer on macroscopic scale. In the future, dynamic rearrangement of the supramolecular building blocks will add an additional level of control not possible with conventional materials, and may allow for the creation of actuators that adapt to specific applications on demand, for example by changing the direction of anisotropic actuation. We anticipate that hybrid materials based on the design principles outlined here will drive the development of superior soft actuators responsive to external stimuli and capable of performing complex mechanical tasks. The grand challenge moving forward is to explore the design of such active matter encoded molecularly to create complex forms of actuation.

## Methods

**Materials**. Acetonitrile, ammonium hydroxide (NH$_4$OH), 2,2'-bipyridine (Bpy), α-bromoisobutyryl bromide, calcium chloride (CaCl$_2$), copper(I) bromide (CuBr), dichloromethane (DCM), di(ethylene glycol) methyl ether methacrylate (DEGMA), diisopropylethylamine (DIEA), N,N-dimethylformamide (DMF), diethylether, ethanol, ethyl α-bromoisobutyrate, fluorescein-O-methacrylate, 1,1,1,3,3,3-hexafluoro-2-propanol (HFIP), methanol, N,N'-methylenebis(acrylamide), 4-methylpiperidine, poly(ethylene glycol) methyl ether methacrylate of M$_n$ 500 (OEGMA$_{500}$), sodium chloride (NaCl), sodium hydroxide (NaOH), sodium persulfate (NaPS), trifluoroacetic acid (TFA), and deuterium oxide (D$_2$O), trifluoroacetic acid-$d_1$, and 4,4-dimethyl-4-silapentane-1-sulfonic acid were purchased from Sigma-Aldrich. Diethanolamine was received from TCI America. All fluorenylmethyloxycarbonyl (Fmoc) protected amino acids were delivered from P3 Biosystems except for Fmoc-ε-Ahx-OH which was obtained from aappTEC. P3 Biosystems also provided the coupling agents 2-(1H-benzotriazol-1-yl)-1,1,3,3-tetramethyluronium hexafluorophosphate (HBTU) and benzotriazol-1-yl-oxytripyrrolidinophosphonium hexafluorophosphate (PyBOP). Carboxytetramethylrhodamine (TAMRA) acid and palmitic acid were obtained from Click Chemistry Tools and Acros Organics, respectively. Triisopropylsilane (TIPS) was purchased from Chem-Impex Int'l INC. DEGMA and OEGMA$_{500}$ were passed over basic alumina to remove inhibitors immediately prior to use. All other chemicals were used as received unless specifically stated otherwise.

**Peptide amphiphile synthesis and purification**. PA molecules were synthesized using standard Fmoc-solid-phase peptide chemistry. Molecules **PA1-3** were synthesized on Rink amide MBHA resin (aappTEC). Fmoc deprotection was performed using 20% 4-methylpiperidine in DMF for 20 min. The resin was then washed with DMF and swollen with DCM. Amino acid couplings were performed with 4 equivalents of protected amino acid, 4 equivalents of HBTU, and 6

equivalents of DIEA in 50% DMF/50% DCM for 2 h. Palmitic acid was coupled to the peptide N-terminus with 8 equivalents of palmitic acid, 8 equivalents of HBTU, 12 equivalents of DIEA in 50% DMF/50% DCM for 4 h. Deprotection and couplings were verified through ninhydrin colorimetric assays (Kaiser test).

Initiator-containing **PA1** was synthesized with a Fmoc-Lys(Mtt)-OH coupled first to the Rink Amide MBHA resin. Deprotection of the 4-methyltrityl (Mtt) group on the lysine ε-amine was performed by swelling the resin in DCM and adding solution of 4% TFA, 5% TIPS and 91% DCM for multiple 5 min washes until yellow color was no longer seen in solution. α-bromoisobutyryl bromide was coupled to the free ε-amine with 4 equivalents α-bromoisobutyryl bromide, 1.1 equivalents PyBOP, and 6 equivalents DIEA in DCM for 4 h. Synthesis of the peptide then continued as described above.

PAs were cleaved from resin by shaking in a solution of 95% TFA, 4% TIPS, 1% water. After cleavage, the solvent was concentrated under reduced pressure and the remaining ~1 mL of solution was precipitated into an excess of cold diethyl ether. The crude product was redissolved at 10 mg mL$^{-1}$ in water with 0.1% NH$_4$OH. This solution was purified using standard preparatory reverse-phase high-performance liquid chromatography (HPLC) techniques on a Shimadzu Prominence instrument equipped with a Phenomenex Gemini NX-C18, 30 × 150 mm column. A mixture of water/acetonitrile containing 0.1% NH$_4$OH was used as eluent. Pure fractions were selected based on the corresponding product signals in electrospray ionization mass spectrometry using direct injection on an Agilent 6520 Q-TOF LC-MS. Organic solvent was removed from selected fractions under reduced pressure before being frozen, lyophilized, and stored at −20 ˚C until further use. Purity of the samples was determined to be > 95% based on the absorption at 220 nm in analytical liquid chromatography-mass spectrometry (LC-MS) using a Agilent 1200 system equipped with a Phenomenex Gemini C18, 1 × 100 mm column with the same Agilent 6520 Q-TOF detector (Supplementary Note 2, Supplementary Figure 1, Supplementary Tables 1-3).

**Tube fabrication**. PA solutions were prepared by dissolving **PAs 1** and **2** in a 1:9 molar ratio in HFIP and sonicating for 1 min. The solvent was removed and the resulting film dissolved at 10 mg/mL in a solution of 50 mM Tris buffer and 150 mM NaCl. The solution was pH-adjusted to pH 7.4 with 1 M NaOH and sonicated for 2 min. The solutions were annealed at 80 ˚C for 30 min and slowly cooled to room temperature over the course of several hours to create a LC solution of PA nanofiber bundles.

Tubes were fabricated using a method adapted from a previously described protocol[26]. Briefly, PA solution was injected via a 25-gauge needle into the annular gap of a 4 mm inner diameter shearing chamber. A 3 mm diameter steel rod was inserted and the chamber fitted onto a 50-mL falcon tube filled with 100 mM CaCl$_2$. The device was loaded onto a modified metal lathe and rotated for 308 ± 5 RPM for 120 s, exerting an estimated rotational shear strain of 97 s$^{-1}$. The translational stage was then retracted slowly, causing CaCl$_2$ solution to flow into the glass tube, immediately gelling the PA suspension. The inner rod was completely removed and the resulting tube was carefully extracted. Laterally aligned samples were prepared in a similar manner but without rotational shear applied; the strain applied in this case is 5-10 s$^{-1}$.

**Hybrid tube polymers**. In a 20-mL scintillation vial, 0.1153 g DEGMA (0.613 mmol, 2850 equiv.) and 0.0161 g OEGMA$_{500}$ (0.032 mmol, 150 equiv.) were mixed with 1.3 mg NBAA (8.43 μmol, 1 wt% of total monomer) and 1.51 mg ascorbic acid (8.6 μmol, 40 equiv.) in 1.5:1 water:methanol (5 mL total). Trace amounts of 4,4-dimethyl-4-silapentane-1-sulfonic acid were added as an internal NMR standard. The solution was mixed and a circumferentially aligned PA tube gel (0.31 mg **PA1**, 2.19 mg **PA2**), made as described above, was added to the solution. The vial was sealed with a rubber septum and degassed with N$_2$ for 15 min. Separately, 3.3 mg CuBr and 8.0 mg Bpy was dissolved in 10 mL methanol in a 50-mL Schlenck flask and degassed with N$_2$. The reaction was initiated by injecting 0.1 mL CuBr/Bpy stock into the reaction vial and placing within a 45 ˚C oil bath for 16 h. 40 μL aliquots were removed at $t = 0$ h and 16 h, and diluted in D$_2$O for $^1$H NMR analysis of monomer conversion. After 16 h, tubes were removed from monomer solution and rinsed repeatedly with milliΩ water to remove residual monomer. Samples were stored in milliΩ water until further analysis.

**Covalent tube polymer**. 0.2273 g DEGMA, 0.0315 g OEGMA$_{500}$, 2.4 mg NBAA, and 2.4 mg NaPS were dissolved in 90 μL milliΩ water and 60 μL ethanol. The mixture was degassed with N$_2$ for 15 min, then placed in a 4 mm diameter cylindrical Teflon mold with a 3 mm central steel post. The mold was sealed and placed in a 67 ˚C water bath for 19 h. The covalent polymer hydrogel was then carefully removed from the Teflon mold, rinsed with milliΩ water, and stored in milliΩ water until further analysis.

**Composite tube polymer**. Water-soluble ATRP initiator 2-bromanyl-N,N-bis(2-hydroxyethyl)-2-methyl propanamide (shown in Supplementary Figure 5) was synthesized by the addition of α-bromoisobutyryl bromide dropwise to a stirred solution of dry diethanolamine in DCM under nitrogen atmosphere overnight. The resulting mixture was extracted with 0.5 M HCl and the organic phase dried under reduced pressure. The initiator was dissolved in a 1:9 molar ratio with **PA2** in HFIP

with sonication, then the solvent was removed and the resulting film redissolved at 1 wt% **PA2** in a solution of 50 mM Tris buffer and 150 mM NaCl. The solution was pH-adjusted to pH 7.4 with 1 M NaOH and sonicated for 2 min. The solutions were annealed at 80 ˚C for 30 min and slowly cooled to room temperature over the course of several hours. Tubes of the annealed solution were prepared as described above, and the tubes subsequently polymerized using the procedure for the hybrid polymers immediately after fabrication.

**3D printed hybrid polymers**. **PA1** and **PA2** were coassembled in a 1:9 molar ratio as described above, but redissolved at a final concentration of 15 mg mL$^{-1}$ before annealing. The annealed PA fibers were printed into 1 cm × 1 cm squares on CaCl$_2$-coated glass substrates using a Hyrel 3D System 30 M printer. Substrates were prepared by first washing glass coverslips with milliΩ water and drying on a hot plate at 70 °C. The clean and dry coverslips were then aerosol spray coated with 0.1 M CaCl$_2$ while still heated at 70 °C, resulting in dispersed 10-50 μm CaCl$_2$ crystals at a surface density of ~0.5 μmol cm$^{-2}$. The PA solution was extruded through a 0.41 mm inner diameter nozzle (Nordson EFD) and gelled immediately upon contact with CaCl$_2$, allowing for string hydrogels to be patterned into squares with varying print paths designed in Slic3r. The tip-to-substrate distance was approximately 200 μm, with a print speed of 10 mm s$^{-1}$ and a flow rate of 0.42 μL s$^{-1}$. Typical gels were ~1mm thick and consisted of 3 or 4 printed layers, with salt introduced via aerosol spray between each subsequent layer. Following printing, the square gels were stored in a hydrated environment to prevent drying before polymerization. The printed PA hydrogels were polymerized using the procedure for the hybrid polymers described above with molar ratios of DEGMA: OEGMA$_{500}$: **PA1**: CuBr: Bpy: ascorbic acid of 14250:750:1:2:5:100. 1 monomer wt % of NBAA crosslinker and 0.5 monomer wt% fluorescein-O-methacrylate were also included in the polymerization solution.

**NMR spectroscopy**. Structural analysis of **PA1** and **PA2** was performed with deuterated trifluoroacetic acid-$d_1$ as the solvent on an Agilent DD 600 MHz w/ HCN cryoprobe ($^1$H) and a Bruker Avance III 500 MHz w/direct cryoprobe ($^{13}$C). Chemical shifts are relative to the solvent signal. Structural assignments (Supplementary Note 1) were performed using $^1$H-$^1$H-gCOSY, $^1$H-$^{13}$C-gHSCQAD, and $^1$H-$^{13}$C-gHMBCAD.

**Anisotropy measurements**. Tubes were sectioned into 7–10 mm segments and placed within a 20 mL glass scintillation vial on a polypropylene base centered on a 21 G needle to prevent translational motion upon heating. The scintillation vial equipped with a temperature probe was filled with sonicated milliΩ water and heated for 30–40 min with a hotplate. A representative temperature ramp is shown in Supplementary Figure 7. Images were taken every 30 s with a Nikon D5100 camera equipped with a Nikon AF-S DX 18–55 mm lens. Images from every 2.5 min were analyzed by measuring length and width at three different tube positions per direction in each image. Distances were normalized to length at $t = 0$. Three independent tubes were measured for each sample condition and averaged. Error bars represent one standard deviation.

**Polarized optical microscopy**. For cross-sectional images the tubes were sectioned into thin segments of approximately 1 mm thickness cut perpendicular to the tube's long axis. The images of the tube's walls were obtained after sectioning along the tube's long axis. The samples were placed in glass bottom 35 mm-dishes (MatTek Corporation, P35G-1.5-14-C) filled with milliΩ water. The samples were then imaged in between two perpendicular light polarizers using a Leica DM750 P instrument in reflection mode with a ×4 magnification using Leica Application Suite V4.2 software. A sample stage allowed for the precise rotation and x, y-positioning of the samples. Cross-sectional images were stitched using the ImageJ plugin tools "MosaicJ" and Grid/Collection Stitching by Preibisch et al[41]. Bright-field images were recorded using the same instrument without the light polarizers.

**Small-angle x-ray scattering (SAXS)**. SAXS measurements were performed at beamline 5-ID-D of the DuPont-Northwestern-Dow Collaborative Access Team (DND-CAT) Synchrotron Research Center at the Advanced Photon Source, Argonne National Laboratory. Liquid samples were prepared at 1 w/v% in 1.5 mm quartz capillaries (Charles Supper), while gel samples were sectioned from tube samples and placed with aqueous solution in an aluminum three-well sample holder with kapton sides. Data was collected using an energy of 17 keV using a CCD detector positioned 245 cm behind the sample. Scattering intensities were recorded within a **q** range of 0.0024 < **q** < 0.40 Å$^{-1}$, where the wave vector **q** is defined as **q** = (4 π/λ) sin(θ/2) where θ is the scattering angle.

For the plot of integrated intensity vs. azimuthal angle "Fit2D" software[42] was used to average the 2D SAXS images of the tubes from at least two separate measurements and the background from a solvent-containing well was subtracted to obtain a 2D image. A radial integration in the **q** range of 0.0024 < **q** < 0.0135 Å$^{-1}$ with an azimuthal angle range of 270 degrees to exclude the beamstop was performed.

**Confocal microscopy**. Fluorescently labeled hybrid tubes were prepared analog to the procedure described above. Fluorescent PA scaffold was prepared by coassembling TAMRA functionalized **PA3** (Supplementary Figure 5) with **PA1** and **PA2** at a molar ratio of 0.01/0.1/0.89. Fluorescein-O-methacrylate was added to the monomer solution at 1 wt% of the monomer fraction to fluorescently label the covalent polymer through copolymerization. For confocal imaging, the tubes were cut along the tubes long axis (tube wall) or sectioned into thin segments of approximately 1 mm thickness cut perpendicular to the tube's long axis (cross-section). The samples were then placed in glass bottom 35 mm-dishes (MatTek Corporation, P35G-1.5–14-C) filled with milliΩ water and visualized using a Nikon A1R confocal laser-scanning microscope equipped with GaAsP detectors.

**Scanning electron microscopy (SEM)**. Hybrid tubes were dehydrated by incubation in a series of ethanol solutions of increasing concentration. Ethanol was subsequently removed by critical point drying (Tousimis Samdri-795). Extra caution was taken to ensure the samples were not heated above the lower critical solution temperature of the polymer material during the exchange. Dehydrated hybrid tubes were mounted on stubs using carbon glue and coated with 21 nm of osmium (Filgen, OPC-60A) to create a conductive sample surface. All SEM images were taken using a Hitachi SU8030 or LEO 1525 instrument operating at an accelerating voltage of 2 kV.

**LC-MS**. The purity of the PAs after HPLC purification was determined by analytical LC-MS using an Agilent 1200 system equipped with a Phenomenex Gemini C18, 1 × 100 mm column, detector Agilent 6520 Q-TOF LC-MS. Gradient: acetonitrile 5% for 5 min at 50 μL min$^{-1}$, 5–95% over 25 min at 50 μL min$^{-1}$ followed by 95% for 5 min at μL min$^{-1}$. Peaks were detected at 220 nm.

**Gel permeation chromatography (GPC)**. Samples were taken by polymerizing hybrids as described above with additional soluble ethyl α-bromoisobutyrate, and aliquots were taken from the aqueous solution. Aliquots were dialyzed against milliΩ water, lyophilized, and reconstituted in tetrahydrofuran (THF). Molecular weights and molecular weight distributions were determined by size exclusion chromatography in THF at 25 °C using two Varian PolyPore 300 × 7.5 mm columns (flow rate 1 mL min$^{-1}$). Detection was performed with multi-angle light scattering (18-angle Dawn Heleos II), viscometer (ViscoStar-II), and differential refractive index (OptiLab T-rex) detectors.

**Cryogenic-transmission electron microscopy (cryo-TEM)**. Cryo-TEM was performed using a JEOL 1230 TEM working at 100 kV accelerating voltage. Samples were plunge frozen using a Vitrobot Mark IV (FEI) vitrification robot at room temperature at 95–100% humidity. 7.5 μL of sample solution (0.1 w/v%, diluted from 1 w/v% immediately before grid preparation) were placed on 300-mesh copper grids with lacey carbon support, blotted, and plunge frozen into liquid ethane. Samples were transferred into a liquid nitrogen bath, and placed into a Gatan 626 cryo-holder through a cryo-transfer stage. Images were acquired using a Gatan 831 CCD camera.

**Microindentation**. Indentation tests were carried out using an axisymmetric probe tack device, consisting of a piezoelectric stepping motor connected in parallel to a load transducer attached to a flat cylindrical indenter with a radius ($R$) of 0.59 mm. Displacement was monitored using an optical sensor with submicrometer sensitivity. All samples with thickness ($h$) of 0.5 mm were fixed on a glass plate with a camera underneath providing optional imaging. The indenter approached the sample with a constant velocity of 10 μm s$^{-1}$ until a specified load, ranging from 10 to 25 mN, was reached. The resulting load ($P$) and displacement ($δ$) data were used to calculate the modulus of the sample. The Young's modulus ($E$) of the sample was calculated by using the relation between the compliance ($C = δ/P$) and the modulus at a specified contact radius ($a$) assuming Poisson's ratio of the sample equals 0.5[43].

$$E = \frac{3P}{8δa}\left(1 + \frac{1.33a}{h} + 1.33\left(\frac{a}{h}\right)^3\right)^{-1} \qquad (4)$$

An average value of $E$ from the above equation from 10–40% strain was used as the modulus. Moduli from multiple indents (3–5 per sample) were averaged to give the reported modulus.

**Tensile testing**. Force-displacement measurements were taken on a Sintech 20 G apparatus with a 2.5 N load cell. Tube samples were removed from water, cut in half longitudinally using a razor blade, and flattened immediately before loading into the apparatus. Gage length and sample dimensions were measured and samples were approximated as a rectangular cross-section. Samples were extended at 5 mm min$^{-1}$ until failure was observed. Data was analyzed using Testworks software to determine modulus and strain at break. Moduli and strain were averaged across multiple samples ($n = 6$–12) to give reported values.

**Calculation of mechanical properties**. The work capacity of the circumferentially aligned tube was calculated using established methods in the actuator field[44],

dividing the work of contraction (kJ) by the mass (kg) of the hybrid material, using the equation below:

$$\text{Work capacity } (\text{kJ kg}^{-1}) = m_\text{w}gΔh/m_\text{h} \qquad (5)$$

Where $m_\text{w}$ is mass of the weights hung from the actuator (kg), $g$ is gravitational acceleration (9.8 m s$^{-2}$), $Δh$ is the displacement of the weight ($m$), and $m_\text{h}$ is the mass of the hybrid (not including water).

The volumetric energy density is calculated as the work during contraction (kJ) divided by the volume of the actuator ($m^3$) using the equation below:

$$\text{Volumetric energy density } (\text{kJ m}^{-3}) = m_\text{w}gΔh/V \qquad (6)$$

Where $V$ = volume of the actuator tube ($m^3$), calculated by multiplying the surface area of the ring by the initial height of the sample.

**Statistical analysis**. Using OriginPro 2017 software, a Welch corrected two sample $t$-test was performed to determine if the normalized width and length of a tube at a given timepoint show statistically significant differences. Statistically significant differences are indicated by $p < 0.05$ (*), $p < 0.01$ (**), and $p < 0.001$ (***).

**Data availability**. The data that support the findings of this study are available from the corresponding author.

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

## Acknowledgements

This work was primarily supported by the Center for Bio-Inspired Energy Science, an Energy Frontier Research Center funded by the U.S. Department of Energy, Office of Science, Basic Energy Sciences under Award # DE-SC0000989. The 3D printing experiments were supported by the Air Force Research Laboratory under agreement number FA8650-15-2-5518. The U.S. Government is authorized to reproduce and distribute reprints for Governmental purposes notwithstanding any copyright notation thereon. The views and conclusions contained herein are those of the authors and should not be interpreted as necessarily representing the official policies or endorsements, either expressed or implied, of Air Force Research Laboratory or the U.S. Government. S.M.C. and A.N.E. acknowledge graduate research fellowships through the National Science Foundation. C.V.S. acknowledges a Feodor Lynen-postdoctoral fellowship through the Humboldt Foundation. Z.A. has received postdoctoral support from the Beatriu de Pinós Fellowship 2014 BP-A 00007 (Agència de Gestió d'Ajust Universitaris i de Recerca, AGAUR), and by Grant #PVA17_RF_0008 from the PVA Research Foundation. N.A.S. was supported by the Department of Defense (DoD), Air Force Office of Scientific Research, through the National Defense Science and Engineering Graduate (NDSEG) Fellowship, 32 CFR 168a and also acknowledges support from Northwestern University International Institute for Nanotechnology through a Ryan Fellowship. X-ray diffraction was performed at the DuPont-Northwestern-Dow Collaborative Access Team (DND-CAT) located at Sector 5 of the Advanced Photon Source (APS). DND-CAT is supported by Northwestern University, E.I. DuPont de Nemours & Co., and The Dow Chemical Company. This research used resources of the Advanced Photon Source, a U.S. Department of Energy (DOE) Office of Science User Facility operated for the DOE Office of Science by Argonne National Laboratory under Contract No. DE-AC02-06CH11357. This work used the following core facilities at Northwestern University: the Peptide Synthesis Core Facility and the Analytical BioNanoTechnology Equipment Core facility (ANTEC) of the Simpson Querrey Institute for peptide synthesis and purification, the Integrated Molecular Structure Education and Research Center (IMSERC) for NMR spectroscopy and GPC, the EPIC facility of Northwestern's NUANCE Center for SEM and DLS, the Biological Imaging Facility (BIF) for TEM, the Central Laboratory for Materials Mechanical Properties (CLaMMP) for tensile testing, and the Center for Advanced Microscopy (CAM) for fluorescence microscopy. The authors thank M. Seniw and B. Jones for the preparation of illustrations, and H. Sai for scientific discussion of SAXS data.

## Author contributions

S.M.C. and C.V.S. synthesized materials, designed and performed experiments, analyzed data, and wrote the manuscript. S.L. and R.J.N. developed and performed computational calculations and wrote the manuscript. Q.W., Z.A., N.A.S., A.N.E., and T.F. performed experiments, analyzed data, and took part in discussions. L.C.P. took part in discussions. I.S. and M.O.d.l.C. supervised research and took part in discussions. S.I.S. wrote the manuscript and supervised the research.
