## [Peer Review File · Nature Communications]

Reviewers' comments:

Reviewer #1 (Remarks to the Author):

This paper reports anisotropic actuation of hydrogel-based actuators. The effect is achieved by using aligned nanofibers, which have been used as template for polymerization of thermoresponsive polymer. Technically paper is well done but novelty is unclear. First, the anisotropic actuation is known and it is also known that mechanical anisotropy is needed to achieve it. The only novelty claimed by authors is absence of complex processing - "We report here on bottom-up molecular design of a macroscopic soft actuator that deforms anisotropically in an aqueous environment without requiring complex processing. The approach consists of the following steps. First oligopeptide-based surfactants which form fibers must be synthesized. These fibers must be aligned in the presence of shear force. Finally, thermoresponsive polymer must be synthesized. Five-step approach illustrated in Figure 1, which required skilled people to be done, can hardly be named simple. Moreover, it is well known that anisotropic particles (fibers and platelets) align in external field (<https://www.nature.com/articles/ncomms2666>). Although, experiments and results are interesting, the authors are encouraged to clearly highlight global novelty of their approach and its advantage. Use of simply new polymers is not sufficient. Claim of simplicity must also be strongly supported.

LCST is about 30°C. What was the reason to cycle temperature between 25°C and 70°C? Is LCST transition not sharp?

Reviewer #2 (Remarks to the Author):

In this report a supramolecular fiber reinforced hydrogel is constructed. The peptide amphiphiles are aligned and determine the level and direction of contraction of the stimulus-responsive composite material. The gels are characterized with a range of techniques and the contraction is described with a model.

The developed material is interesting because of its controlled deformation behavior initiated by a self-assembled hierarchical fiber network. This latter part brings a new element to the field of fiber reinforced gels, and as such makes it potentially interesting for Nature Comm. However, a number of comments first have to be addressed.

First of all, the supramolecular fiber reinforcement is the most novel element of this manuscript. However, the authors have not explored this property in this paper. In situ fiber disassembly and possible reassembly would add a layer of control not possible with more conventional materials. One possible way to test this is to perform a shearing experiment on the gels. This would lead to a yield stress when the fibers break, and a restoration effect when they are reformed. The mechanical properties of the gels are not studied, and this should be performed anyhow (moduli, stress-strain curves, behavior under shear).

The authors have studied only the 10% grafted composites. What are the effects on actuation using different graft ratios?

There is a difference between the grafted and non-grafted composites. Can the authors exclude a phase separation in the latter case?

Reviewer #3 (Remarks to the Author):

The manuscript "Covalent-Supramolecular Hybrid Polymers as Muscle-Inspired Anisotropic Actuators" by S. M. Chin et al investigates a new interesting bottom-up design of macroscopic temperature-driven actuators of pure organic origin. The authors utilize muscle-inspired principles of self-assembly of structurally different components to achieve robust materials exhibiting

reproducible anisotropic actuation of fibers and 3D printed hybrid sheets. The subject of the manuscript is of obvious interest of a wide audience of the journal as nature-inspired materials and actuators in particular attracts considerable attention of researchers during the last decade. The advantages of the approach is its simplicity, incorporation of synthetic materials, good level of property control and capability of further expansion of the applied principles for further improvement or modification of material properties. Another strength of the manuscript is application of several experimental characterization techniques in combination with a theoretical/molecular modeling approach to achieve molecular level understanding of the most important elements of structural design of the materials. Overall the results are exciting, well analyzed and presented clearly. The manuscript can be published after a minor revision upon which the following points could be addressed:

1. It would be useful, if experimentally feasible, to be able to estimate the degree of actual crosslinking of the nanofibers and measure the force associated with contraction/expansion. The former is among the most important parameters determining viscoelastic properties of the material and the latter is one of the desired properties of any actuator.
2. The predicted effect of polymer grafting density Fig.5C is rather small, so it could be moved to the supporting information. Instead authors could include Fig.S14 showing the strong effect of the hydrophobic fraction in grafted polymer or PEG length Figure S15. It may be an important result for material design.

Response to Reviewers' Comments and Text Modifications of

"Covalent-Supramolecular Hybrid Polymers as Muscle-Inspired Anisotropic Actuators"

By Stacey M. Chin, Christopher V. Synatschke, Shuangping Liu, Rikkert J. Nap, Qifeng Wang, Nicholas A. Sather, Zaida Álvarez, Alexandra N. Edelbrock, Timmy Fyrner, Liam C. Palmer, Igal Szleifer, Monica Olvera de la Cruz, Samuel I. Stupp

Reviewer #1 Comments	Authors' Response and Text Modifications
This paper reports anisotropic actuation of hydrogel-based actuators. The effect is achieved by using aligned nanofibers, which have been used as template for polymerization of thermoresponsive polymer. Technically paper is well done but novelty is unclear. First, the anisotropic actuation is known and it is also known that mechanical anisotropy is needed to achieve it. The only novelty claimed by authors is absence of complex processing - "We report here on bottom-up molecular design of a macroscopic soft actuator that deforms anisotropically in an aqueous environment without requiring complex processing. The approach consists of the following steps. First oligopeptide-based surfactants which form fibers must be synthesized. These fibers must be aligned in the presence of shear force. Finally, thermoresponsive polymer must be synthesized. Five-step approach illustrated in Figure 1, which required skilled people to be done, can hardly be named simple. Moreover, it is well known that anisotropic particles (fibers and platelets) align in external field (https://www.nature.com/articles/ncomms2666). Although, experiments and results are interesting, the authors are encouraged to clearly highlight global novelty of their approach and its advantage. Use of simply new polymers is not sufficient. Claim of simplicity must also be strongly supported.	We appreciate the comment by Reviewer 1 concerning the novelty of our work. As mentioned in the introduction, several different materials that have been previously reported are also capable of performing anisotropic actuation, such as refs 16-22. We may not have been particularly clear on the main differences that our system exhibits, compared to these previously reported materials. We hope to better highlight the novelty of our work with the following remarks: As mentioned by Reviewer 2, our system uses a "self-assembled hierarchical fiber network" to provide directional information in our system. The use of self-assembling, organic molecules in a bio-inspired approach for such a purpose is novel; most other systems use inorganic fillers to create anisotropic composite materials. We have added wording to the introduction to highlight this: ("In supramolecular materials, it is possible to introduce a directional bias of non-covalent assemblies using external forces,¹⁵ however, very few examples of using such materials as actuators have been reported." page 3 lines 18–20). As it is a very appropriate example, we have also included the suggested reference into the introduction ("A different approach to anisotropic actuation has been the use of organic-inorganic composite materials to introduce directionality in the microstructure.²¹", Page 4, line 3-5). The phrase "directional information" (page 4 line 23) is used here to include the contributions from both the mechanical anisotropy and the brush effect from the

grafting procedure; an approach that has not been explored in a controlled manner. Our work clearly shows that our hybrid is a superior actuator with stronger anisotropic actuation (Fig. 2a) compared to the “simple” composite material (Fig. 4b) that only features mechanical anisotropy, but no grafted polymer chains (“We found that composite samples do exhibit anisotropic actuation, shrinking to 85% of their original length and 92% of their original width. However, this actuation is of considerably lower magnitude than that of the previously described circumferentially aligned hybrid samples (Fig. 2a).” Page 12, lines 14-17).

We acknowledge the difficulty in defining simplicity of fabrication. From our point of view, the shear stress applied to align these materials (maximum 1-3 Pa, estimating flow at the edge of a pipe at the maximum strain rate for fabrication of 100 s⁻¹ with a viscosity of 0.01-0.03 Pa/s) are lower than those required in other common actuators (multiple Tesla magnetic fields or MPa stresses). We have altered the text in a number of places to make our claims about fabrication simplicity more clear to the reader:

- Page 4, line 9-11: “It remains of great interest in materials science to design anisotropic actuators based on intrinsically ordered systems containing supramolecular and covalent polymers that align under weak shear forces”
- Page 4, line 13-15: “We report here on the bottom-up molecular design of a macroscopic soft actuator that deforms anisotropically in an aqueous environment without requiring high temperature or high stress processing, which are typically needed in such systems.²⁴”
- Page 5, line 1-3: “As described below, the required steps for bottom-up assembly of this muscle-inspired hierarchically ordered actuator utilize easily applied shear using simple benchtop procedures.”

LCST is about 30°C. What was the reason to cycle temperature between 25°C and 70°C? Is LCST transition not sharp?

While the LCST of the PEGMA polymer in solution is sharp, occurring at 35°C, we expect that the mechanical contraction may be slower due to the need for water diffusion through the entire material thickness (Matsuo, E. S.; Tanaka, T., Kinetics of discontinuous volume–phase transition of gels. *The Journal of Chemical Physics* 1988, 89 (3), 1695-1703). Therefore, we cycled to more extreme temperatures in order to get the maximum amount of contraction. Holding at a constant temperature just above the LCST would have a very similar response; however, our experimental setup is not configured for isothermal heating.

To clarify, in the revised manuscript we have added DLS measurements of the PA-polymer compared to polymer alone and PA alone in solution to Supplemental Fig. 3c to show the LCST transition (SI Page 7, Fig. S3), as well as noted the LCST transition in the main text: “The random copolymer exhibits the expected lower critical solution temperature (LCST) of 35°C by DLS (**Supplementary Fig. 3**).³⁰” Page 7, line 9-10.

Reviewer #2 Comments	Authors' Response and Text Modifications
In this report a supramolecular fiber reinforced hydrogel is constructed. The peptide amphiphiles are aligned and determine the level and direction of contraction of the stimulus-responsive composite material. The gels are characterized with a range of techniques and the contraction is described with a model. The developed material is interesting because of its controlled deformation behavior initiated by a self-assembled hierarchical fiber network. This latter part brings a new element to the field of fiber reinforced gels, and as such makes it potentially interesting for Nature Comm. However, a number of comments first have to be addressed.	We thank the reviewer for the comments and suggestions. We have addressed the issues as described below.
First of all, the supramolecular fiber reinforcement is the most novel element of this manuscript. However, the authors have not explored this property in this paper. In situ fiber disassembly and possible reassembly would add a layer of control not possible with more conventional materials. One possible way to test this is to perform a shearing experiment on the gels. This would lead to a yield stress when the fibers break, and a restoration effect when they are reformed.	We agree that the supramolecular fiber reinforcement is a novel element. However, in the current material configuration it is likely that the nanofibers are not fully dynamic, as we “freeze” the alignment through the gelation with calcium. This ionic crosslinking likely reduces the mobility of the nanostructures (shown with similar systems previously reported by the Stupp group https://pubs.acs.org/doi/abs/10.1021/jacs.7b02969) and prevents the suggested in situ disassembly and reassembly. In the future, however, we plan to fully explore the capabilities of the supramolecular backbone through a more reversible fixation of the shear alignment, such as pH. This may also require altering the β-sheet forming region of the peptide amphiphile in order to tune the strength of the supramolecular interactions, which goes beyond the scope of our current molecular design. Unfortunately, we are unable to perform the suggested shearing experiments with conventional rheological fixtures due to the unconventional tubular geometry of our material. Furthermore, we expect that any data obtained would be difficult to deconvolute, as there would be contributions

	from both the supramolecular architecture and covalent crosslinks. We have included a statement discussing this future work in the revised manuscript: “In the future, dynamic rearrangement of the supramolecular building blocks will add an additional level of control not possible with conventional materials, and may allow for the creation of actuators that adapt to specific applications on demand, for example by changing the direction of anisotropic actuation.” Page 20, lines 10-12.
The mechanical properties of the gels are not studied, and this should be performed anyhow (moduli, stress-strain curves, behavior under shear).	The compressive moduli of the hybrid materials was measured using microindentation, showing a fourfold increase in radial compressive modulus (16.7 kPa) vs the initial supramolecular scaffold (4.2 kPa) (added to Page 7, line 21–Page 8, line 1). We have also completed new tensile testing experiments (Supplementary Fig. 13) to further examine the mechanical properties and added this information to the revised manuscript. We saw an increased strain at break and corresponding lower Young’s modulus for the circumferentially aligned samples over the longitudinally aligned samples. (“Tensile experiments show a difference in the Young’s modulus perpendicular and parallel to the nanofiber alignment axis (50.3 ± 39 kPa vs 263.3 ± 179 kPa), indicating the nanofiber orientation does affect the material mechanics (Supplementary Fig. 13).” Page 12, line 5-8).
The authors have studied only the 10% grafted composites. What are the effects on actuation using different graft ratios?	Supplementary Fig. 16 shows the predicted effect of grafting density (directly related to grafting ratio) based on molecular modeling. This shows that we expect some change in the degree of actuation depending on grafting ratio — however, there may be an upper limit to the amount of actuation due to steric limitations. We have added additional text explaining what we would expect to see experimentally in the Supplemental Information (“We expect some increase in actuation with increased grafting above the experimental grafting density of 0.07 chains/nm²; however, at high grafting

	densities the changes are minimal due to increased sterics preventing polymer mobility close to the nanofiber surface. “ P. 25, paragraph 4), as well as in the main text (“variations in grafting density in the range of that used experimentally result in only minor changes in contraction (Supplementary Fig. 16a).” Page 18, line 8-9. We have also added to the revised manuscript preliminary experiments with varying initiator density in the Supplemental Information. These initial experiments show similar amounts of contraction, but different rates of contraction. We note these differences, however a more detailed analysis of the kinetics is beyond the scope of the work and will be explored in the future. (“This is corroborated experimentally in Supplementary Fig. 17, where increasing the grafting density by increasing the mol% of initiator in the supramolecular coassembly. The overall degree of contraction remains similar across three cases; however, the kinetics of the contraction appear to change. These kinetics will be explored in future work.” Page 25, paragraph 4).
There is a difference between the grafted and non-grafted composites. Can the authors exclude a phase separation in the latter case?	Laser-scanning confocal microscopy of both the hybrid materials and non-grafted composites is shown in the supplemental information (Figure S6, page 9). These materials appear to be very similar with regards to high colocalization and features on the micron-scale, so we believe there is no large-scale phase separation of the supramolecular and covalent components. Additional clarifying text has been added to the revised manuscript: “These composite samples show similar integration of the covalent polymer component throughout the material (Supplementary Fig. 6).” P. 12, line 13-14.

Reviewer #3 Comments	Authors' Response and Text Modifications
The manuscript "Covalent-Supramolecular Hybrid Polymers as Muscle-Inspired Anisotropic Actuators" by S. M. Chin et al investigates a new interesting bottom-up design of macroscopic temperature-driven actuators of pure organic origin. The authors utilize muscle-inspired principles of self-assembly of structurally different components to achieve robust materials exhibiting reproducible anisotropic actuation of fibers and 3D printed hybrid sheets. The subject of the manuscript is of obvious interest of a wide audience of the journal as nature-inspired materials and actuators in particular attracts considerable attention of researchers during the last decade. The advantages of the approach is its simplicity, incorporation of synthetic materials, good level of property control and capability of further expansion of the applied principles for further improvement or modification of material properties. Another strength of the manuscript is application of several experimental characterization techniques in combination with a theoretical/molecular modeling approach to achieve molecular level understanding of the most important elements of structural design of the materials. Overall the results are exciting, well analyzed and presented clearly. The manuscript can be published after a minor revision upon which the following points could be addressed:	We thank the reviewer for the positive comments. We have addressed the issues raised as described below.
1. It would be useful, if experimentally feasible, to be able to estimate the degree of actual crosslinking of the nanofibers and measure the force associated with contraction/expansion. The former is among the most important parameters determining viscoelastic properties of the material and the latter is one of the desired properties of any actuator.	While it is extremely difficult to determine experimentally, the degree of crosslinking is estimated at one link per 75 monomer units based on initial molar ratios (page 5, line 23). We expect both inter- and intrafiber crosslinking to occur. The following text has been added to the revised manuscript: "Crosslinking is expected to occur both within polymer chains on the same nanofiber as well as between nanofibers." Page 6, line 1-2.

	In order to determine the force exerted by the actuator, work experiments and calculations of the work capacity and volumetric energy density were done in the revised manuscript (Shown in Supplementary Fig. 9). Additional text referring to these experiments was added to the main text (“The actuation of these materials is sufficiently strong to perform work (Supplementary Fig. 9), with samples able to lift up to 380 times their dry weight with a work capacity of 0.629 kJ/kg and volumetric energy density of 5.656 kJ/m³.” Page 8, lines 12-14).
2. The predicted effect of polymer grafting density Fig.5C is rather small, so it could be moved to the supporting information. Instead authors could include Fig.S14 showing the strong effect of the hydrophobic fraction in grafted polymer or PEG length Figure S15. It may be an important result for material design.	We have taken the reviewer’s advice and replaced Fig. 5c with Fig. S14 (page 16), showing the stronger effect of the hydrophobicity on the thermoresponse. We have also added additional text alluding to its potential role in materials design: “We found that the strategy to control the thermal response is to vary the hydrophobic component of the polymer, which can drastically change the transition temperatures (Fig. 5c). This could be an important handle for the design of further actuating materials.” Page 18, line 18-20.

REVIEWERS' COMMENTS:

Reviewer #2 (Remarks to the Author):

The authors have adequately answered the comments raised. Although not all suggested experiments have been performed, the argumentation against these experiments by the reviewers sounds plausible to me. With the amendments made I feel the paper is now suitable for publication.

Reviewer #3 (Remarks to the Author):

In the revised version of the manuscript authors addressed all my comments and concerns and did a very good job addressing comments of other reviewers. I think manuscript was improved and made stronger upon the revision and can be accepted in its current form.

Editorial Note: Reviewer #3 looked over Reviewer #1's comments in the last round and the authors' response, and found that all concerns were satisfactorily addressed. However 'easy preparation' is subjective and it is advised to tone down this claim.

RESPONSE TO REVIEWERS' COMMENTS:

REVIEWERS' COMMENTS:

Reviewer #2 (Remarks to the Author):

The authors have adequately answered the comments raised. Although not all suggested experiments have been performed, the argumentation against these experiments by the reviewers sounds plausible to me. With the amendments made I feel the paper is now suitable for publication.

Response: We thank the reviewer for the positive comment.

Reviewer #3 (Remarks to the Author):

In the revised version of the manuscript authors addressed all my comments and concerns and did a very good job addressing comments of other reviewers. I think manuscript was improved and made stronger upon the revision and can be accepted in its current form.

Response: We are grateful for the reviewer's positive comments.

Editorial Note: Reviewer #3 looked over Reviewer #1's comments in the last round and the authors' response, and found that all concerns were satisfactorily addressed. However 'easy preparation' is subjective and it is advised to tone down this claim.

Response: We agree that the term "easy preparation" is subjective and that it may have been used too strongly in our original version of the paper. We did adjust the language of the manuscript in the revised version. In this third version, we have removed the term 'easily' from Page 15, line 1 (Full markup of Tracked Changes) of the main text (now reading "Our material can be aligned with weak shear forces..."). The only remaining reference related to "easy preparation" is on Page 5, line 2: "As described below, the required steps for bottom-up assembly of this muscle-inspired hierarchically ordered actuator utilize easily applied shear using simple benchtop procedures." We believe this particular statement is justified with the qualifications given and would like to keep it as is in the manuscript. We hope this is OK.